# GRK specificity and Gβγ dependency determines the potential of a GPCR for arrestin-biased agonism
Edda S. F. Matthees [1,6], Jenny C. Filor [1,6], Natasha Jaiswal[1,2], Mona Reichel [1], Noureldine Youssef [1], Giulia D'Uonnolo[3,4], Martyna Szpakowska[3], Julia Drube [1], Gabriele M. König[5], Evi Kostenis [5], Andy Chevigné [3], Amod Godbole[1] & Carsten Hoffmann [1] ✉

G protein-coupled receptors (GPCRs) are mainly regulated by GPCR kinase (GRK) phosphorylation and subsequent β-arrestin recruitment. The ubiquitously expressed GRKs are classified into cytosolic GRK2/3 and membrane-tethered GRK5/6 subfamilies. GRK2/3 interact with activated G protein βγ-subunits to translocate to the membrane. Yet, this need was not linked as a factor for bias, influencing the effectiveness of β-arrestin-biased agonist creation. Using multiple approaches such as GRK2/3 mutants unable to interact with Gβγ, membrane-tethered GRKs and G protein inhibitors in GRK2/3/5/6 knockout cells, we show that G protein activation will precede GRK2/3-mediated β-arrestin2 recruitment to activated receptors. This was independent of the source of free Gβγ and observable for Gs-, Gi- and Gq-coupled GPCRs. Thus, β-arrestin interaction for GRK2/3-regulated receptors is inseparably connected with G protein activation. We outline a theoretical framework of how GRK dependence on free Gβγ can determine a GPCR's potential for biased agonism. Due to this inherent cellular mechanism for GRK2/3 recruitment and receptor phosphorylation, we anticipate generation of β-arrestin-biased ligands to be mechanistically challenging for the subgroup of GPCRs exclusively regulated by GRK2/3, but achievable for GRK5/6-regulated receptors, that do not demand liberated Gβγ. Accordingly, GRK specificity of any GPCR is foundational for developing arrestin-biased ligands.

Upon agonist stimulation, a G protein-coupled receptor (GPCR) induces structural changes in the bound heterotrimeric G protein to induce G protein-dependent signalling[1]. GPCR kinases (GRKs) play a critical role in regulating this signalling, as they are the major class of kinases to phosphorylate active GPCRs to initiate receptor desensitization by enhancing arrestin recruitment[2–4]. Due to initial reports demonstrating that unwanted side effects of opioid treatment were reduced in β-arrestin2 knockout mice[5], multiple studies have since been aimed to introduce bias between G protein- or β-arrestin-mediated GPCR pathways[6–9]. In general, biased signaling describes the capability of a ligand at a given receptor to preferentially trigger one signaling pathway over another when compared to a reference ligand[10,11]. In a ternary complex of a ligand, a receptor and a transducer, any of the three components can theoretically be the cause of an observed signaling bias[12]. Currently, the mechanistic aspects of biased agonism are described from two perspectives. The first aspect would be the structural component, which is inherent to possible conformations of a GPCR, coupling differentially to distinct cellular signaling pathways, and can be ideally controlled by selective ligands[13,14]. The second component contains the respective transducer and is often unknown and encoded with the factor τ, which is used to express biased factors for ligands and contains the cellular contributors of biased signaling[15]. However, which overarching mechanism for GPCRs generally determines the shift in balance between G protein and β-arrestin pathways remains unclear.

With the aim of unravelling one such cellular component in this mechanism, we utilized CRISPR/Cas9-edited HEK293 cells devoid of GRK2/3/5/6 (ΔQ-GRK cells) to observe that GPCRs show GRK selectivity

[1]Institut für Molekulare Zellbiologie, CMB – Center for Molecular Biomedicine; Universitätsklinikum Jena, Friedrich-Schiller-Universität Jena, Hans-Knöll-Straße 2, D-07745 Jena, Germany. [2]Department of Internal Medicine, Section of Gerontology and Geriatric Medicine, Section of Molecular Medicine, Wake Forest University School of Medicine, Winston-Salem, NC, USA. [3]Immuno-Pharmacology and Interactomics, Department of Infection and Immunity, Luxembourg Institute of Health (LIH), 29 rue Henri Koch, L-4354 Esch-sur-Alzette, Luxembourg. [4]Faculty of Science, Technology and Medicine, University of Luxembourg, Esch-sur-Alzette, Luxembourg. [5]Molecular, Cellular and Pharmacobiology Section, Institute for Pharmaceutical Biology, University of Bonn, Nussallee 6, D-53115 Bonn, Germany. [6]These authors contributed equally: Edda S. F. Matthees, Jenny C. Filor. ✉e-mail: carsten.hoffmann@med.uni-jena.de

to mediate β-arrestin recruitment[16]. Of particular importance was the finding that some receptors relied solely on GRK2/3-mediated regulation, whereas others did not show any preference among the four ubiquitously expressed GRKs (GRK2/3/5/6-regulated) (Table 1)[16]. Recent studies additionally indicated the existence of a third category of only GRK5/6-dependent receptors (Table 1), which so far only includes intrinsically β-arrestin-biased receptors, naturally unable to trigger G protein activation[17,18]. Interestingly, an increasing amount of evidence suggests that receptor phosphorylation by different GRK isoforms leads to distinct outcomes, generally summarized in the field by the barcode hypothesis[19–22].

This intriguing GRK selectivity led us to revisit the basic molecular mechanisms of how GRKs are stabilized at the plasma membrane to interact with GPCRs and to evaluate this as a crucial and central molecular mechanism contributing to biased signalling. As previously reported, the cytosolic GRK2/3 rely on free Gβγ subunits to translocate to the membrane at the site of active receptor[23–27]. Only after this Gβγ-mediated GRK2/3 recruitment, can the receptor be phosphorylated to enhance β-arrestin2 recruitment[28]. Conversely, the membrane localization of GRK5/6 is independent of free Gβγ proteins[29–31]. Utilizing ΔQ-GRK cells, we systematically evaluated that indeed Gβγ interaction is crucial for GRK2/3-mediated β-arrestin recruitment to Gs-, Gi- and Gq-coupled GPCRs whereas interaction with Gα is negligible.

Our findings could have significant impact on studies of biased agonism for G protein *versus* β-arrestin pathways since our data imply that all GRK2/3-mediated β-arrestin effects are indeed G protein-(βγ)-dependent and only GRK5/6-mediated effects on β-arrestin are G protein-independent. In this study, we outline a theoretical framework of how GRK dependence on free Gβγ can influence a GPCR's potential in biased agonism.

## Results

### GRK2-Gβγ binding is essential to recruit β-arrestin2 to b2AR

To investigate the G protein dependency of GRK2-mediated β-arrestin2 recruitment to activated GPCRs, we used a variety of GRK2-mutants with low binding affinity towards the G protein subunits and a quadruple knockout cell line devoid of ubiquitously expressed GRK isoforms 2/3/5 and 6 (ΔQ-GRK cells)[16]. As depicted in Fig. 1a, these mutations in GRK2 disrupt its interaction with Gαq (GRK2-D110A)[24], its interaction with Gβγ (GRK2-R587Q)[32,33] or both (double mutation at D110A and R587Q in GRK2)[28]. The mutant constructs were assessed for similar expression levels using Western blot (Supplementary Fig. 1)[34]. We also created CAAX-tagged versions of GRK2 and the above mutants[35] to localize the cytosolic GRKs permanently to the plasma membrane (Supplementary Fig. 2) and thus overwrite the requirement on binding to active G protein subunits to be recruited to an activated receptor.

First, we investigated the dependency of β-arrestin2 on interaction between activated G proteins and GRKs when recruited to the prototypical Gs-coupled beta-2 adrenergic receptor (b2AR). Previously, we demonstrated that each GRK2/3/5 and 6 can enhance recruitment of β-arrestin to b2AR upon agonist stimulation[16]. Here, we re-introduced all GRK2 mutants mentioned above in combination with Nanoluciferase (NLuc)-tagged b2AR

and Halo-tagged β-arrestin2 in ΔQ-GRK cells and systematically monitored b2AR–β-arrestin2 interaction via NanoBRET upon stimulation with the agonist isoproterenol (Iso).

For data visualization, we plotted the normalized net BRET change as concentration-response curves (Fig. 1b–g) as well as in bar graphs for the highest agonist concentration to compare the effect of the different GRK2 versions (Fig. 1h, i, detailed statistical results are provided in Supplementary Table 2). The GRK2-D110A mutant has been described to disrupt binding of GRK2 specifically with Gαq[24,26]. As anticipated our findings show that this mutant did not negatively influence β-arrestin2 recruitment to the Gs-coupled b2AR as compared to wild type (WT) GRK2 (Fig. 1b, h). The introduction of the CAAX motif to the WT GRK2 or the GRK2-D110A did not affect its ability to mediate β-arrestin2 recruitment to the b2AR (Fig. 1c, i, Supplementary Fig. 3, Supplementary Table 3). These findings imply that the interaction between GRK2 and the Gα subunit is secondary. Interestingly, disruption of GRK2 binding to free Gβγ subunits using the GRK2-R587Q mutant did significantly reduce β-arrestin2 recruitment to amplitudes seen in the condition without endogenous GRK2/3/5/6 expression (Fig. 1d, h). The localization of this GRK2-R587Q mutant on the plasma membrane rescued β-arrestin2 recruitment to similar amplitude as seen for the WT GRK2-CAAX (Fig. 1e, i) further highlighting that removing the dependence of GRK2 on its interaction with activated Gβγ subunits can restore receptor phosphorylation and hence β-arrestin recruitment. In line with these findings, the double mutant (GRK2-D110A,R587Q) also drastically reduced β-arrestin2 recruitment compared to WT GRK2 (Fig. 1f, h) whereas removing the dependence on activated Gβγ subunits by using the plasma membrane-localized GRK2 double mutant restored β-arrestin2 recruitment as seen for the WT GRK2-CAAX (Fig. 1g, i).

With this observation that GRK2-regulated β-arrestin2 recruitment to a Gs-coupled receptor would always be preceded by free Gβγ subunits, we investigated whether this is independent of the Gβγ source.

### GRK2-Gβγ interaction is necessary to mediate β-arrestin2 recruitment to the muscarinic acetylcholine M2 and M5 receptors

To this aim, we then tested the prototypical Gi-coupled muscarinic M2 acetylcholine receptor (M2R), which has been shown by us to be dependent on only GRK2/3 to mediate β-arrestin recruitment[16]. We tested the dependency of β-arrestin2 recruitment to this receptor on GRK–G protein interactions by using similar systematic approaches and mutants as described above (Fig. 2a, b). As seen for the b2AR (Fig. 1), the disruption of the binding of GRK2 to free Gβγ subunits by using the GRK2-R587Q mutant significantly reduced β-arrestin2 recruitment to the M2R, an effect which was shared also with the double mutant (Fig. 2a, Supplementary Table 4). Targeting GRK2 to the membrane using the CAAX-tag led to a reduced dynamic change of β-arrestin2 recruitment to this receptor (Supplementary Fig. 4, Supplementary Table 5). Nevertheless, utilizing the GRK2-CAAX version allowed the M2R to recruit β-arrestin2 by surpassing the need of free Gβγ subunits as shown by the GRK2-R587Q-CAAX mutant (Fig. 2b).

To test the dependency of β-arrestin2 on GRK–G protein interaction for a Gq-coupled receptor, we investigated the muscarinic M5 acetylcholine receptor (M5R), which we have shown to be GRK2/3-regulated with respect to β-arrestin recruitment[16]. As seen for the previous receptors, disruption of GRK2 binding to active Gβγ subunits significantly reduced β-arrestin2 recruitment (Fig. 2c, Supplementary Table 6). Interestingly, disruption of GRK2 binding to the Gαq subunit did not reduce β-arrestin2 recruitment (Fig. 2c). This is in line with previous reports that GRK2-Gαq interaction is secondary to GRK-Gβγ interaction[35]. These findings now performed without an endogenous GRK background clearly unravel the role of free Gβγ subunits on the ability of GRK2 to regulate receptors. As seen for the other receptors, surpassing the need of free Gβγ subunits by using CAAX-tagged GRK2, the recruitment of β-arrestin2 to M5R was restored to the same levels of WT GRK2-CAAX (Fig. 2d). Similarly to the M2R, introducing the CAAX motif to the GRK2 reduced the agonist-dependent dynamic change of mediated β-arrestin2 recruitment comparted to WT GRK2

## Table 1 | Overview of classified GRK2/3-, GRK2/3/5/6-, GRK5/6-regulated GPCRs in current literature

| GRK2/3 | GRK2/3/5/6 | GRK5/6 |
|---|---|---|
| M2R[16] | AT1R[16,47] | C5aR2[17] |
| M4R[16] | C5aR1[16,17] | ACKR3[18] |
| M5R[16] | CCR2[17] | |
| MOP[16,53] | M3R[16] | |
| | PTH1R[16] | |
| | V2R[16] | |
| | b2AR[16] | |
| | GLP1R[54] | |

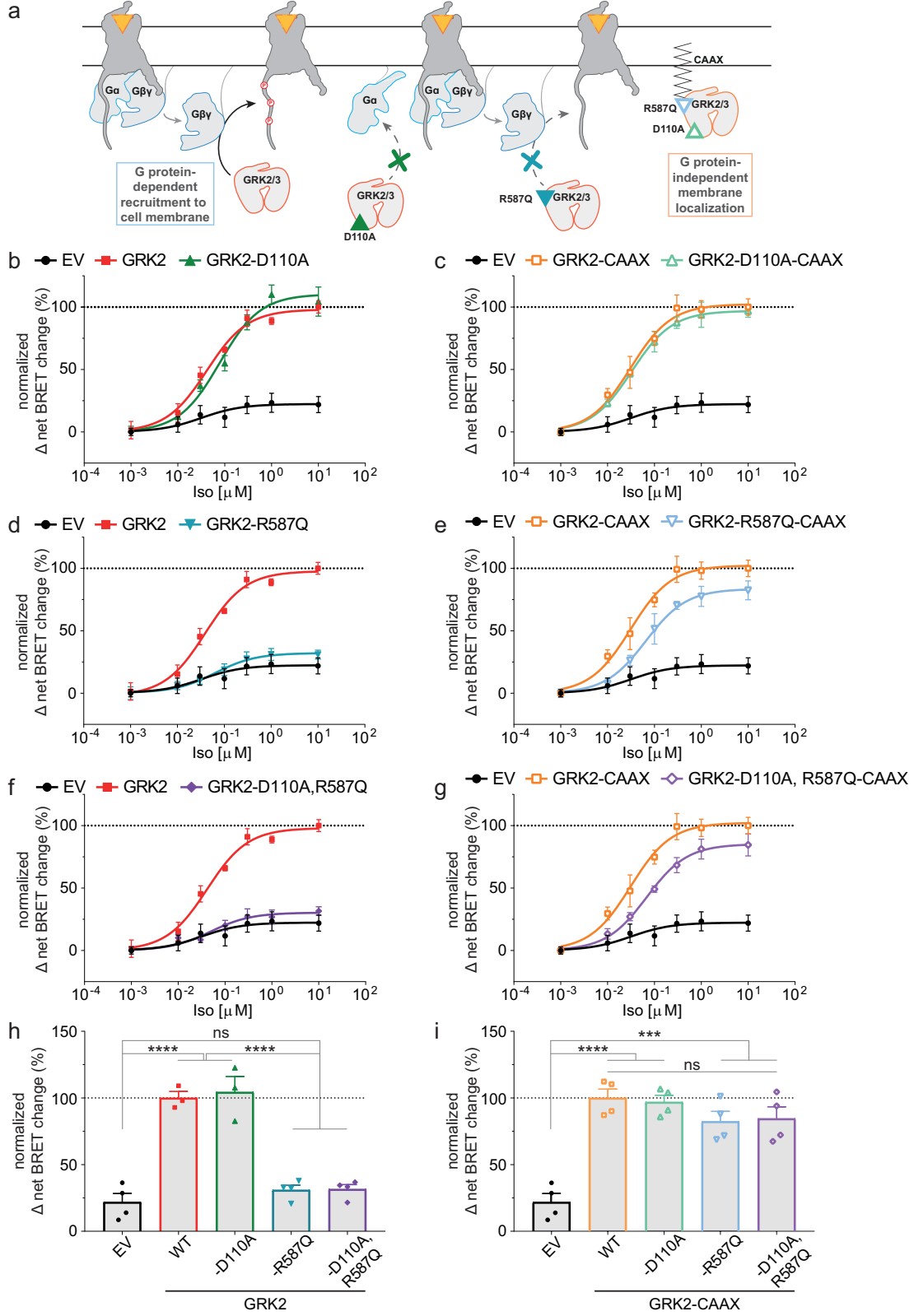

(Supplementary Fig. 5, Supplementary Table 7), indicating that the functional receptor interaction might be different for the membrane-tethered GRK2 with these GPCRs as opposed to the b2AR (Supplementary Fig. 3). Nevertheless, the relative observation held true for all the compared receptors.

In summary, the data with Gs-, Gi- and Gq-coupled receptors point out a general mechanism in which free Gβγ subunits play a critical role in translocating cytosolic GRK2 to the plasma membrane. This Gβγ dependency was generally circumvented by introducing the membrane localization motif (CAAX) to the GRK2.

**Fig. 1 | GRK2-mediated β-arrestin2 recruitment to the beta-2 adrenergic receptor (b2AR) is dependent on the membrane localization of the GRK. a** Schematic representation of the utilized GRK mutants D110A (interrupting the GRK2/3–Gα interaction), R587Q (interrupting the GRK2/3–Gβγ interaction) and the double mutant (D110A, R587Q), also as versions with a CAAX box to localize GRK2/3 to the plasma membrane independent of the G protein interaction. **b–g** Isoproterenol (Iso)-induced Halo-Tag-β-arrestin2 recruitment to b2AR-NanoLuciferase (NLuc) in GRK2/3/5/6-depleted quadruple knockout HEK293 (ΔQ-GRK) cells in absence of the ubiquitously expressed GRKs (empty vector (EV)-transfected) and in presence of wild type (WT) GRK2 or either GRK2-D110A (**b**), GRK2-R587Q (**d**) or GRK2-D110A,R587Q (**f**). The same experiment was performed with the

corresponding GRK2-CAAX versions (**c, e, g**). All data are shown as Δ net BRET change in percent of $n = 4$ (except WT GRK2 (**a**) and GRK2-D110A (**b**) which are $n = 3$) independent experiments ± SEM, normalized to the maximum response with GRK2 (**b, d, f**) or GRK2-CAAX (**c, e, g**). The curves in absence of ubiquitously expressed GRKs (EV), GRK2 and GRK2-CAAX are shown multiple times to allow direct comparisons. **h, i** Normalized BRET data of the highest stimulation of **b–g** are displayed as bar graphs and statistical differences were tested using one-way ANOVA, followed by a Tukey's test (*ns* not significant; *$p < 0.05$; **$p < 0.01$; ***$p < 0.001$; ****$p < 0.0001$). Detailed statistical results are provided in Supplementary Table 2.

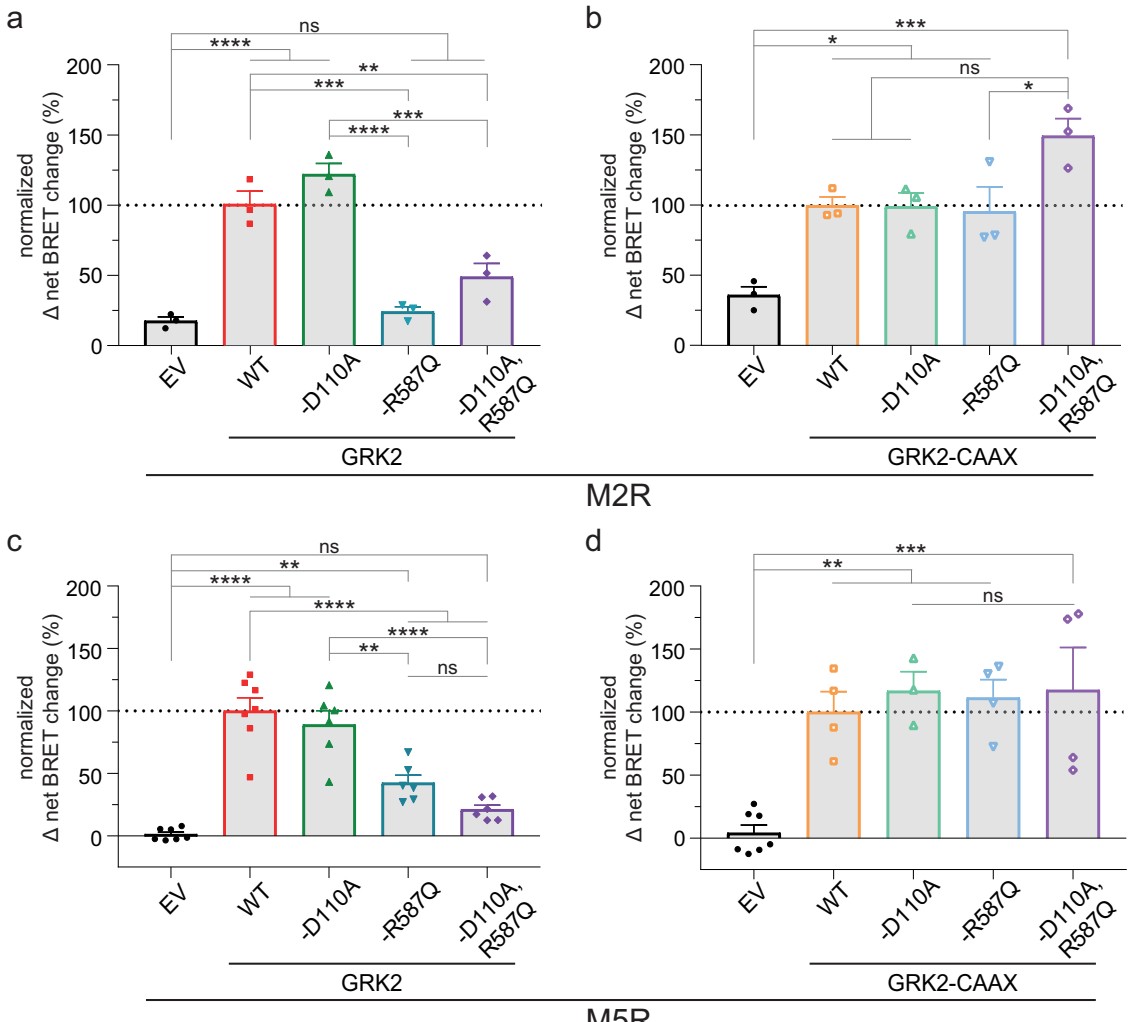

**Fig. 2 | β-arrestin2 recruitment to the muscarinic M2 and M5 acetylcholine receptors (M2R, M5R) is dependent on the ability of the GRK to be recruited to the membrane. a–d** Halo-Tag-β-arrestin2 recruitment to M2R-NLuc (**a, b**) or M5R-NLuc (**c, d**) was measured in ΔQ-GRK cells in absence of GRKs (EV-transfected) and in presence of WT GRK2, GRK2-D110A, GRK2-R587Q, GRK2-D110A,R587Q or their respective membrane-tethered versions via a CAAX box (**b, d**). Normalized Δ net BRET change (%) upon stimulation with 100 μM of Acetylcholine (ACh) is shown for M2R (**a, b**) of $n = 3$ and for M5R (**c, d**) of $n = 4$ (except M5R: EV and WT

GRK2 of $n = 7$; GRK2-D110A, GRK2-R587Q and GRK2-D110A,R587Q of $n = 6$; GRK2-D110A-CAAX of $n = 3$) independent experiments ± SEM, normalized to GRK2 (**a, c**) or GRK2-CAAX (**b, d**). The data measured in absence of GRKs (EV) is shown in both graphs each (**a, b** for M2R; **c, d** for M5R) for direct comparison. Statistical differences were tested using one-way ANOVA, followed by a Tukey's test (*ns* not significant; *$p < 0.05$; **$p < 0.01$; ***$p < 0.001$; ****$p < 0.0001$). Detailed statistical results are provided in Supplementary Table 4 (M2R) and Supplementary Table 6 (M5R).

## GRK3 is similarly dependent on membrane-recruitment to phosphorylate GPCRs as family member GRK2

Next, we tested the dependency of β-arrestin2 on GRK3–G protein interactions since GRK3 is the other cytosolic subfamily member and yet less investigated kinase in the field of receptor regulation. In an analogous approach to GRK2, we utilized the mutants and their CAAX-tethered

versions, as described above (Supplementary Fig. 6, Supplementary Table 8). We obtained similar results with GRK3 and its Gαq-(D110A) or Gβγ-(R587Q) interaction mutants, as well as the double mutant (Fig. 3a, c, e, Supplementary Table 9–11). The introduction of the CAAX motif reduced the ability of the GRK to mediate β-arrestin2 recruitment to these receptors (Supplementary Fig. 7, Supplementary Table 12), similar to what we

observed for GRK2. In contrast to GRK2, the GRK3-D110A, R587Q-CAAX mutant version generally interfered with the extent of mediated β-arrestin2 recruitment stronger compared to the other GRK3-CAAX versions (Fig. 3b, d, f). This potentially indicates some differences between the two closely related GRK isoforms. Still, the findings for the cytosolic GRK3–Gβγ interaction mutants strongly indicate how receptors, which are solely regulated by GRK2/3, are largely dependent on agonist-dependent G protein rearrangement.

### GRK2/3-mediated β-arrestin2 recruitment is reduced by competitive inhibition of GRK2/3-Gβγ-interaction

In this study, we show that GRK2/3-mediated β-arrestin2 recruitment is dependent on the interaction of GRK2/3 with Gβγ by mutation of the Gβγ interaction site at GRK2/3 (Figs. 1–3). To investigate this principle using a different approach, we utilized the C-terminal domain of GRK2, referred to as bARK-CT, as a described competitive inhibitor for GRK2/3–Gβγ interaction[32,36,37]. The identified GRK2–Gβγ interaction site R587[32,33] is included in the bARK-CT peptide (Fig. 4a). We investigated the influence of bARK-CT co-expression on GRK2/3-mediated β-arrestin2 recruitment to the M5R (Fig. 4b). Of the investigated receptors, the M5R was chosen since it is described as a GRK2/3-regulated GPCR, also under endogenous expression levels of the ubiquitously expressed GRKs unlike the M2R[16]. The b2AR was omitted as it is regulated by GRK2/3/5 and 6[16]. Based on this, the experiment was performed at endogenous expression levels of GRKs in HEK293 control cells. To test the influence of bARK-CT on GRK2/3-mediated β-arrestin2 recruitment 0.5 µg or 1 µg of bARK-CT was transfected as indicated in addition to NLuc-tagged M5R and Halo-tagged β-arrestin2, as described above.

Increasing amounts of transfected bARK-CT led to a gradual reduction in the dynamic range of β-arrestin2 recruitment (Fig. 4b, Supplementary Fig. 8, Supplementary Table 13). This indicated that not just the mutation of the Gβγ interaction site in GRKs, but also the competitive inhibition of endogenous GRK2/3 by bARK-CT leads to a reduction of GRK2/3-mediated β-arrestin2 recruitment. These findings indicate that utilization of bARK-CT, which is a known inhibitor of Gβγ signaling, as a competitive GRK2/3 inhibitor would also affect β-arrestin2 recruitment.

### Utilization of a guanine nucleotide dissociation inhibitor diminished GRK2/3-mediated β-arrestin2 recruitment and receptor internalization

To further strengthen our observation, we investigated the influence of a different class of inhibitory compound, FR900359, which was published to be a Gq-specific guanine nucleotide dissociation inhibitor[38], on direct β-arrestin2 recruitment to the GRK2/3-regulated, Gq-coupled M5R and the subsequent receptor translocation to early endosomes (Fig. 5, Supplementary Fig. 9). According to our forth-put hypothesis, inhibition of heterotrimeric G protein dissociation should lead to a reduction of free Gβγ, hence decreasing GRK2/3 recruitment to the activated receptor and subsequent phosphorylation-dependent β-arrestin2 recruitment (Fig. 5a). Indeed, we found a significantly reduced β-arrestin2 recruitment to the M5R in presence of the Gq-inhibitor FR900359 when endogenous GRKs are expressed (Fig. 5b, detailed statistical results are provided in Supplementary Table 14). To confirm that this ultimately also reduces receptor internalization, we measured M5R translocation to early endosomes utilizing an early endosome-tethered fluorophore (FYVE-mNeonGreen)[39–41] (Fig. 5c). Utilization of FR900359 significantly reduced M5R internalization in cells with endogenous GRK expression to comparable levels as in absence of the ubiquitously expressed GRKs (ΔQ-GRK cells) (Fig. 5d, detailed statistical results are provided in Supplementary Table 14). These findings demonstrate that established guanine nucleotide dissociation inhibitors like FR900359 also greatly affect β-arrestin2 recruitment and β-arrestin2-supported functions such as receptor internalization for GRK2/3-dependent GPCRs. Taken together, these data further highlight the dependency of cytosolic GRK2/3 on free Gβγ subunits to translocate to the membrane and initiate β-arrestin recruitment via receptor phosphorylation.

## Discussion

In this study, we investigated the impact and the necessity of the interaction between Gβγ and cytosolic GRK2/3 as a cellular mechanism for GRK2/3 translocation to and stabilization at the plasma membrane. We specifically studied the effect of this interaction on GPCR–β-arrestin complex formation. To this end, we utilized a number of previously described mutations in GRK2/3, which selectively interrupt either binding to Gβγ[32,33] or Gαq[24] or both[28] and combined these mutations with a CAAX motif to tether the cytosolic GRKs to the plasma membrane independently of the disrupted Gβγ interaction. We performed these measurements systematically in a HEK293 knockout cell line devoid of endogenous GRK2/3/5/6 background[16]. Using this unique combination, we could show that the ability of GRK2 to mediate b2AR–β-arrestin2 interaction strongly depends on Gβγ binding (Fig. 1). Next, we demonstrated that this critical dependency of GRK2 on Gβγ binding is independent of the G protein coupling specificity of GPCRs as it is conserved for a Gs-, Gi- and Gq-coupled receptor (Figs. 1, 2). This indicates that we did not observe specificity of the Gβγ source as seen for GIRK-channel activation via Gβγ of Gi proteins[42] but rather a general impact of Gβγ. Analogously, we showed that this mechanism is also conserved for GRK3 across all investigated GPCRs (Fig. 3), albeit the clarity of these results interestingly appears to be somewhat less pronounced than for GRK2. In other words, GRK2/3 subfamily-mediated β-arrestin recruitment to GPCRs is severely determined by the ability of these GRKs to bind to free Gβγ subunits and hence, Gβγ availability. To test this observation with different approaches, we additionally investigated this in cells expressing the endogenous wild type GRK complement using bARK-CT, as a known Gβγ inhibitor, or FR900359, an established guanine nucleotide dissociation inhibitor. Utilization of bARK-CT or FR900359 led to an inhibition of β-arrestin recruitment to the GRK2/3-regulated M5R (Figs. 4, 5). Further, the utilization of the specific Gq inhibitor FR900359 led to a strong reduction of M5R translocation to early endosomes (Fig. 5). Thus, unless other, yet unknown mechanisms of membrane recruitment substitute for this Gβγ interaction, G protein activation and GRK2/3-mediated β-arrestin recruitment are inseparably intertwined. When reviewing the literature, we found no prior evidence that this was comprehensively analyzed before in the absence of endogenous GRK background. However, it has been stated that a component of GRK2/3 recruitment requires GPCR signaling to liberate free Gβγ[43,44], although the degree of dependency was unknown.

Our findings presented in this article suggest that the GRK2/3 interaction with Gβγ is a fundamental mechanism and has far-reaching consequences for the global effort of creating β-arrestin-biased agonists: we propose that the GRK isoform requirement of a GPCR to recruit β-arrestin determines the potential of creating biased agonists promoting β-arrestin recruitment without activating G proteins. Hence, we would like to broadly classify GPCRs based on their GRK selectivity and connect this classification to a consequential strategy to create β-arrestin-biased agonists (Fig. 6).

According to this GRK selectivity classification, we group GPCRs into GRK2/3-, GRK2/3/5/6- or GRK5/6-dependent receptors (Fig. 6). The creation of pure β-arrestin-biased agonists would be mechanistically challenging for GRK2/3-regulated GPCRs, such as the M2R or M5R, as the GRK2/3-mediated β-arrestin effects are indeed Gβγ-dependent. Partial agonists could lead to G protein-biased signaling if the subsequently available Gβγ is not sufficient to mediate efficient GPCR phosphorylation by GRK2/3. However, the thresholds for these mechanisms remain unknown. If activation of and phosphorylation by GRKs exhibit some degree of amplification, it could be also imaginable that weak G protein activation may still result in comparable β-arrestin recruitment levels.

Further, receptors that are solely GRK5/6-regulated have intrinsically the highest possibility of obtaining β-arrestin-biased agonists since such agonists would not rely on the added necessity of a mechanistic G protein activation to induce phosphorylation and arrestin binding. Until recently, the only two GRK5/6-regulated seven transmembrane receptors described in the literature were atypical ones, which are intrinsically β-arrestin-biased, naturally lacking G protein coupling[17,18]. In other words, these receptors do

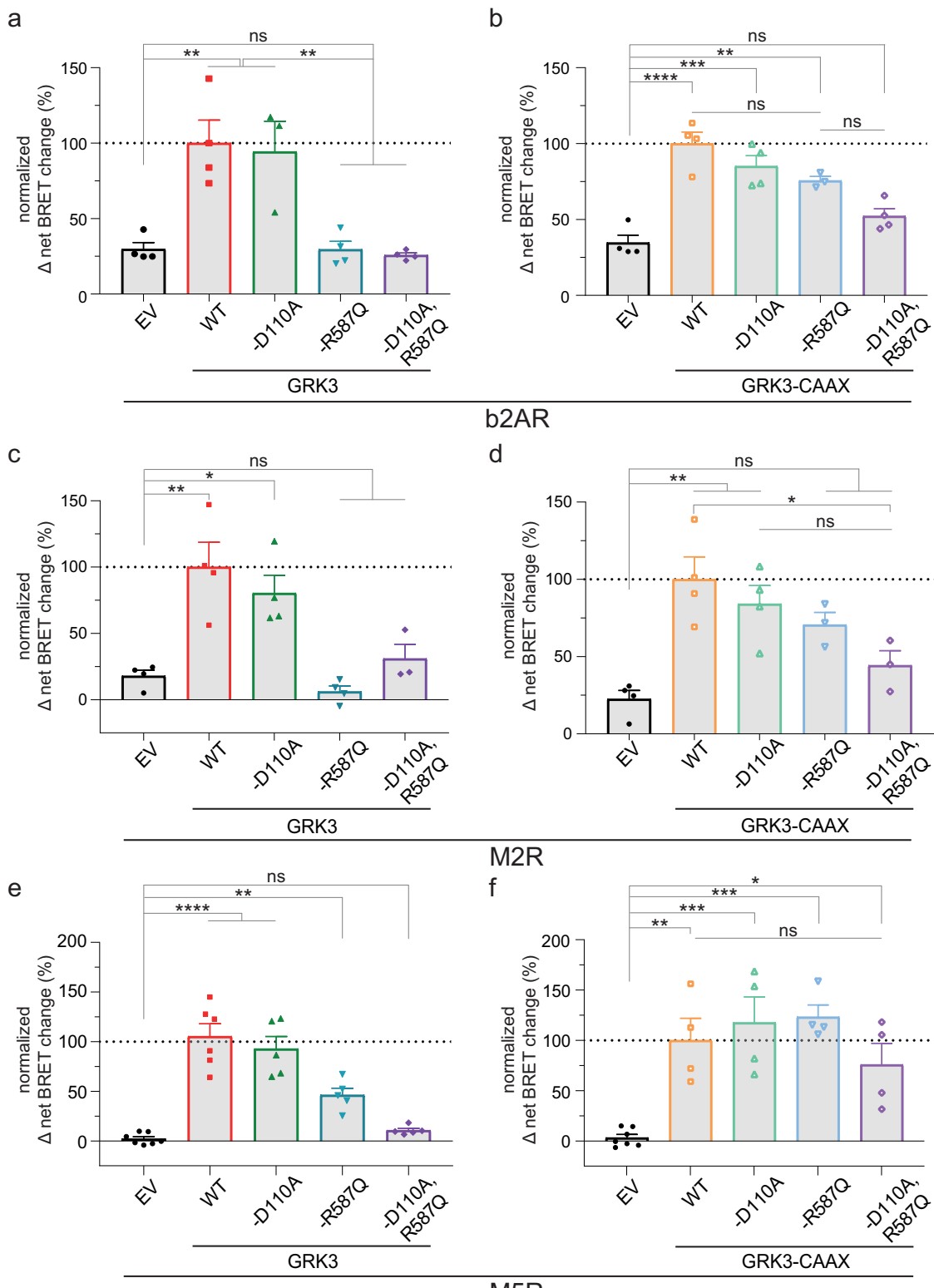

**Fig. 3 | GRK3-mediated β-arrestin2 recruitment to b2AR, M2R and M5R displayed a similar dependency as GRK2 on the ability of the GRK to be recruited to the membrane. a–f** Halo-Tag-β-arrestin2 recruitment to b2AR-NLuc (**a, b**), M2R-NLuc (**c, d**) or M5R-NLuc (**e, f**) was measured in ΔQ-GRK cells analogous to GRK2 in absence of GRKs (EV-transfected) and in presence of WT GRK3, GRK3-D110A, GRK3-R587Q, GRK3-D110A,R587Q or their respective membrane-tethered versions via a CAAX box (**b, d, f**). Normalized Δ net BRET change (%) upon stimulation with 10 μM Iso (**a, b**) or 100 μM of ACh (**c–f**) is shown for b2AR (**a, b**) and M2R (**c, d**) of $n = 4$ (except b2AR: GRK3-D110A and GRK3-R587Q-CAAX of $n = 3$; and M2R: GRK3-

D110A,R587Q, GRK3-R587Q-CAAX and GRK3-D110A,R587Q-CAAX of $n = 3$), for M5R GRK3 mutants (**e**) of $n = 5$ (except EV of $n = 7$ and WT GRK3 of $n = 6$) and for all GRK3-CAAX constructs of M5R (**f**) of $n = 4$ independent experiments ± SEM, normalized to GRK3 (**a, c, e**) or GRK3-CAAX (**b, d, f**). The data measured in absence of GRKs (EV) is shown in both graphs respectively (**a, b** for b2AR; **c, d** for M2R; **e, f** for M5R) for direct comparison. Statistical differences were tested using one-way ANOVA, followed by a Tukey's test (*ns* not significant; *$p < 0.05$; **$p < 0.01$; ***$p < 0.001$; ****$p < 0.0001$). Detailed statistical results are provided in Supplementary Table 9 (b2AR), Supplementary Table 10 (M2R) and Supplementary Table 11 (M5R).

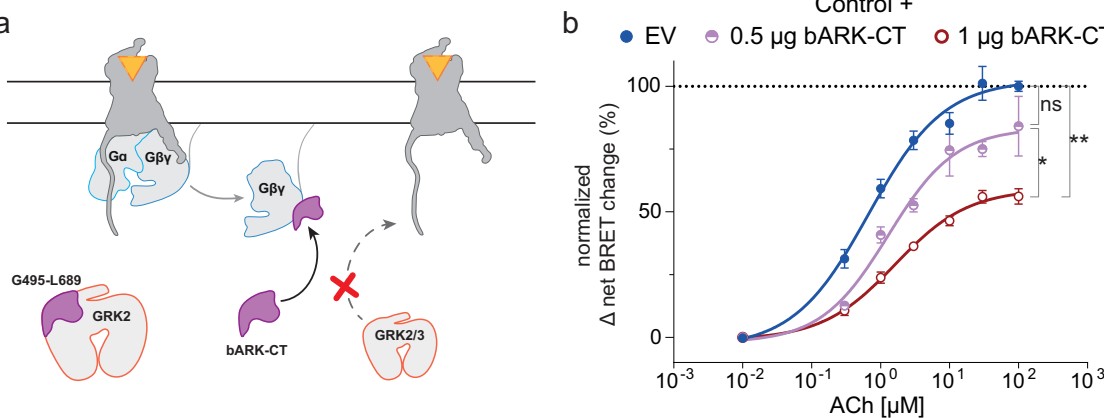

**Fig. 4 | bARK-CT reduces β-arrestin2 recruitment to the muscarinic M5 acetylcholine receptor (M5R). a** Schematic representation of the bARK-CT-mediated mechanism as an inhibitor of GRK2/3 recruitment to the M5R. The bARK-CT fragment of GRK2 includes the GRK2-Gβγ interaction site (R587) and therefore competes with GRK2/3 for the binding. **b** Halo-Tag-β-arrestin2 recruitment to M5R-NLuc was measured in CRISPR/Cas9 HEK293 control cells, expressing all GRKs at endogenous levels, in absence (empty vector (EV)-transfected) or presence of different co-transfected amounts of bARK-CT (as indicated). Normalized Δ net BRET change (%) upon stimulation with the indicated concentrations of Acetylcholine (ACh) is shown of $n = 4$ (except 0.5 µg bARK-CT which is $n = 3$) independent experiments ± SEM, normalized to EV. Statistical differences were tested using one-way ANOVA, followed by a Tukey's test (*ns* not significant; $^*p < 0.05$; $^{**}p < 0.01$). Detailed statistical results are provided in Supplementary Table 13. Comparison of basal and stimulated values can be found in Supplementary Fig. 8.

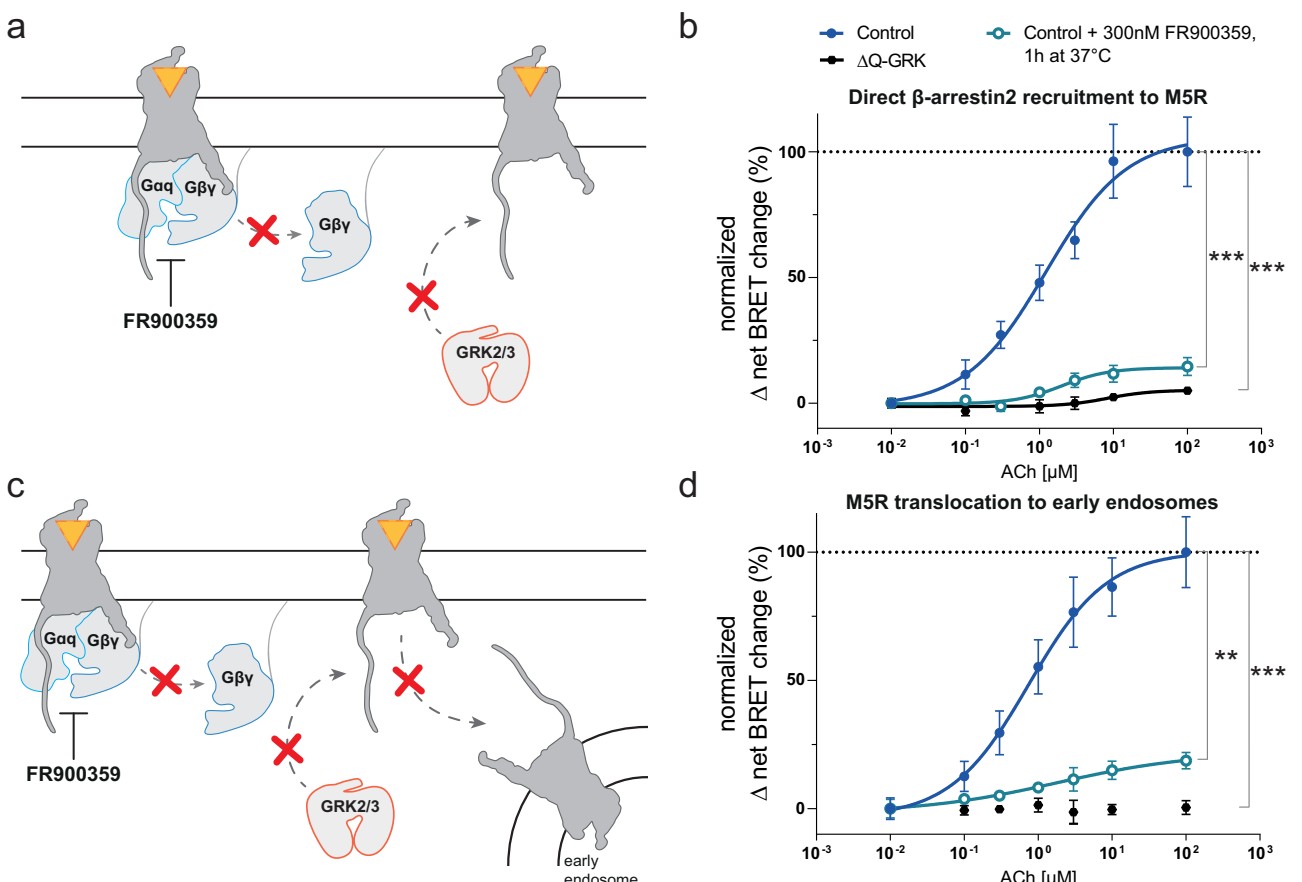

**Fig. 5 | FR900359 reduces β-arrestin2 recruitment to the muscarinic M5 acetylcholine receptor (M5R) and receptor translocation to early endosomes. a** Schematic representation of FR900359-mediated reduction of free Gβγ as a guanine nucleotide dissociation inhibitor. **b** Halo-Tag-β-arrestin2 recruitment to M5R-NLuc was measured in CRISPR/Cas9 HEK293 control cells (Control), expressing all GRKs at endogenous levels, in absence or presence of 300 nM FR900359 or in quadruple GRK2/3/5/6 knockout cells (ΔQ-GRK). **c** Schematic representation of FR900359-mediated reduction of free Gβγ and subsequent inhibition of M5R internalization. **d** M5R-NLuc translocation to early endosomes (FYVE-NeonGreen) as a measure of internalization was assessed in Control cells in absence or presence of 300 nM FR900359 or in ΔQ-GRK cells. All data are shown as normalized Δ net BRET change (%) upon stimulation with the indicated concentrations of Acetylcholine (ACh) of $n = 3$ independent experiments ± SEM, normalized to Control without FR900359 incubation. Statistical differences were tested using one-way ANOVA, followed by a Tukey's test (*ns* not significant; $^{***}p < 0.001$; $^{****}p < 0.0001$). Detailed statistical results are provided in Supplementary Table 14. Δ net BRET fold changes over time and the comparison of basal and stimulated values can be found in Supplementary Fig. 9.

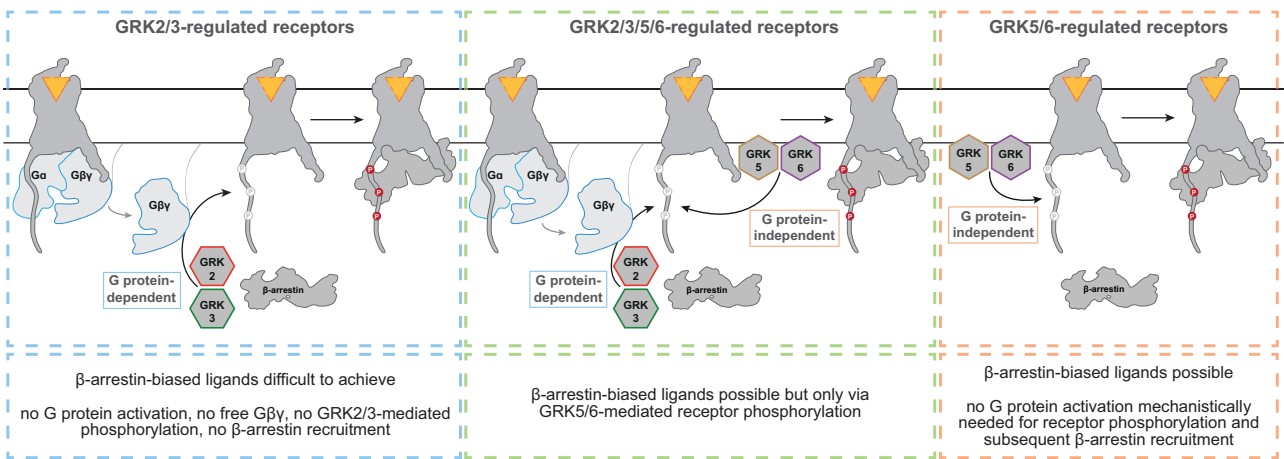

**Fig. 6 | The GRK-dependency of each GPCR determines its potential in biased agonism.** GPCRs can be grouped based on the GRKs involved in their regulation into GRK2/3-dependent, GRK5/6-dependent or GRK2/3/5/6-dependent receptors. As the membrane-localization of GRK2/3 is mediated via the interaction with Gβγ, the phosphorylation of the receptor and hence, the β-arrestin recruitment, are in fact G protein-dependent. Therefore, it will likely be mechanistically unattainable or difficult to achieve for this group of receptors to create β-arrestin-biased ligands that do not activate G proteins, because the phosphorylation by GRK2/3 is dependent on the availability of free Gβγ-subunits. This is not the case for GRK5/6-regulated receptors, as these GRKs are already membrane-tethered and not dependent on Gβγ for the recruitment to this receptor group. For receptors that are found to be GRK2/3/5/6-regulated, β-arrestin-biased ligands would convey their effects only via GRK5/6-induced receptor phosphorylation.

not activate G proteins and therefore there would be no free Gβγ subunits available to mediate GRK2/3 recruitment to these receptors. It has been shown for the atypical chemokine receptor ACKR3 that GRK2/3-mediated phosphorylation can be induced by overexpression of Gβγ and delivery of free Gβγ in the receptor vicinity via hetero-dimerization with the G protein-activating CXCR4, if stimulated by CXCL12 as a shared agonist for both receptors[45]. In this case, the same ligand induced active conformations of the receptors, which might have enabled GRK2/3 to phosphorylate also the atypical ACKR3 even though the recruitment to the membrane was mediated via CXCR4. Recently, the GPR35 has also been described as a GRK5/6-regulated GPCR[46]. Since this receptor has been shown to be Gi-, G12/13-coupled, this implies that additional factors might contribute to GRK specificity in this group of receptors.

Finally yet importantly, as β-arrestin recruitment to GRK2/3/5/6-regulated GPCRs is partly G protein-dependent (GRK2/3) and partly independent (GRK5/6), the creation of β-arrestin-biased agonists targeting GRK2/3/5/6-regulated GPCRs is generally possible. However, these would only mediate β-arrestin effects linked to GRK5/6 phosphorylation while GRK2/3 effects would be lacking since the absence of G protein activation would not deliver free Gβγ. Strong support for this hypothesis can be found in the literature. Two independent studies demonstrated clearly in GRK-knockout cell lines that the primarily Gq-coupled angiotensin-II (AngII) type 1 receptor (AT1R) is GRK2/3/5/6-regulated[16,47]. In Gq-knockout cells, this GRK specificity profile was shifted towards exclusive dependency on GRK5/6 for the balanced ligand AngII[47]. In this study, the same observation was made when wild type HEK293A cells were incubated with the chemical Gq inhibitor YM-254890. Using GRK family knockout cells, the authors also showed that the established β-arrestin-biased agonist TRV027 only mediates β-arrestin recruitment via GRK5/6 and no longer via GRK2/3 at the AT1R, as TRV027-mediated β-arrestin recruitment was only measurable in GRK2/3 knockout cells and not if GRK5/6 were knocked out[47]. We would argue that this originates from the lack of G protein activation and subsequent absence of free Gβγ to recruit GRK2/3 to the membrane. This is further supported by a recent study demonstrating that also for the GRK2/3/5/6-regulated b2AR, the GRK specificity shifted towards GRK5/6 exclusively in Gs-knockout cells[48]. Furthermore, this publication clearly demonstrates that the cellular outcomes, e.g. gene regulation, differ strongly between the WT condition and the Gs-knockout cells, where no G protein activation and therefore no GRK2/3-mediated regulation of the b2AR is possible. Hence, when designing β-arrestin-biased agonists one should keep

in mind that the GRK5/6-mediated β-arrestin downstream effects might differ substantially from the functional outcome facilitated by GRK2/3/5/6-recruited β-arrestin[19,48,49].

Of note, we previously observed for some receptors a residual β-arrestin recruitment independently of GRKs[16]. This might open the possibility of GRK-independent bias for specific receptors. However, distinct β-arrestin conformational changes were measured in presence or absence of GRKs, indicating different functional outcomes[50]. Future studies should aim to illuminate which β-arrestin-supported effects are carried out by GRK2/3 or GRK5/6 phosphorylation or independently from GRKs. Our current findings in this conceptual work allowed this simple, previously impossible classification of GPCRs based on their GRK selectivity. This classification of receptors into these three categories in combination with structural knowledge and other described factors influencing bias might be a key step in understanding if, which and how biased agonists will be possible.

In summary, we propose an understanding of what contributes to β-arrestin-biased agonism and how this biased agonism is highly dependent on which GRKs can regulate the activated receptor. To this aim, we pieced together all the available information on GRK2/3–Gβγ interactions and GRK2/3/5/6 cellular locations. We then used hitherto unavailable tools such as the recently published GRK knockout cell line in combination with established BRET assays to decipher the effect of Gβγ subunits on individual GRKs. Moreover, on a broader scale, these cell lines are a perfect starting point to characterize novel GPCRs, which would be candidates for screening biased agonists. One would simply need to characterize this novel GPCR of interest and understand which GRKs regulate the arrestin binding. This information would be the key and the first hint of possibilities to create a biased agonist. Since the assay show-cased in this study uses NLuc-tagged receptors expressed in these cell lines (all available on request), one could simply use this assay to expand the repertoire of tested agonists and investigate whether the screened agonists can change GRK selectivity. Employing cell lines stably expressing the receptor of interest, this would have the potential to be a high throughput screening method to asses GRK selectivity for a myriad of agonists. Using these tools, we are putting forth a straightforward yet until now underappreciated systematic understanding of the mechanism and the key players dictating biased agonism: the availability of free Gβγ subunits and the selectivity of receptors towards a specific set of GRKs. Ultimately, our findings hope to convey the importance of evaluating whether a receptor exhibits a "GRK bias" to assess optimal strategies for inducing G protein- or β-arrestin-mediated cellular responses.

## Methods

### Cloning and construct origin

The transfected GRK constructs and receptors were of human origin and β-arrestin2 was of bovine origin. The NanoLuciferase (NLuc) and Halo-Tag genes were obtained from Promega and the plasmids expressing b2AR-, M2R- and M5R-NLuc and Halo-β-arrestin2 have been described before[16]. The bARK-CT construct was provided by Professor Silvio Gutkind. Plasmids expressing human full-length GRK2 and GRK3 in pcDNA3 have been described in Drube et al., 2022[16] and plasmids expressing GRK2/3-D110A and GRK2/3-R587Q have been described and characterized in Jaiswal, 2023[34]. The mutant constructs were assessed for similar levels via Western blot analysis[34]. The GRK2-D110A/R587Q double mutant was created by using GRK2-D110A as the template and 5'GGAGATCTTCGCCTCAT ACATCATGAAGGAGCTGCTGG as forward primer and 5'CCAGC AGCTCCTTCATGATGTATGAGGCGAAGATCTCC as reverse primer. GRK3 D110A/R587Q double mutant was created using GRK3 R587Q as the template and 5'TTCCCCAACCAGCTCGAGTGGC as the forward primer and 5'GCCACTCGAGCTGGTTGGGGAA as the reverse primer to introduce D110A. GRK2/3-CAAX constructs were generated by inserting the CAAX overhangs using Gibson assembly to GRK2/3 WT, D110A, R587Q and D110A/R587Q. The generated constructs were validated by sequencing at Eurofins Genomics GmbH. The pcDNA3 backbone was used as a control and is referred to as the empty vector (EV). The mNeonGreen-GRK constructs were generated through the insertion of dsDNA strings (GeneArt) encoding GRK2/3 WT, -D110A, -R587Q and -D110A, R587Q with or without a following sequence encoding the CAAX motif (NPPDESGPCCMSCKCVLS), downstream the mNeonGreen coding sequence, using In-Fusion Cloning technology (Takara Bio)[51,52].

### Cell culture

CRISPR/Cas9-generated HEK293 knockout cells of GRK2/3/5/6 (ΔQ-GRK), GRK2/3 (ΔGRK2/3) or GRK2 (ΔGRK2) and CRISPR/Cas9 HEK293 control cells (Control) with unaltered GRK expression[16] were cultured at 37 °C with 5% $CO_2$ in Dulbecco's modified Eagle's medium (DMEM; Sigma-Aldrich, D6429), complemented with 10% fetal calf serum (Sigma-Aldrich, F7524) and 1% penicillin and streptomycin mixture (Sigma-Aldrich P0781). Cells were passaged every 3–4 days and regularly checked for infections with mycoplasma using the LONZA MycoAlert mycoplasma detection kit (LT07-318).

### Western blot

The following antibodies were used for the detection of GRK2/3 and their mutants: mouse anti-GRK2 (Santa Cruz, sc-13143), rabbit anti-GRK3 (Cell signaling technology, 80362), mouse anti-actin (Sigma-Aldrich, A5441), HRP-conjugated goat anti-mouse (SeraCare, 5220-0341) and HRP-conjugated goat anti-rabbit (SeraCare, 5220-0336).

GRK2 knockout cells (GRK2 Western blot) or GRK2/3 knockout cells (GRK3 Western blot) were washed once with PBS and subsequently lysed with RIPA buffer (1% NP-40, 1 mM EDTA, 50 mM Tris pH 7.4, 150 mM NaCl, 0.25% sodium deoxycholate) at room temperature for 15 min. The lysates were centrifuged at 10,000 g for 15 min at 4 °C and the protein amount was estimated using the BCA protein estimation kit (Thermo Fisher Scientific, 23225). Sample loading buffer was added to each cell lysate followed by denaturation at 95 °C for 5 min. 10 µg of total protein was loaded onto each lane of 10% polyacrylamide gels and following electrophoretic separation, was transferred to nitrocellulose membranes. After the membranes were blocked with TBST (supplemented with 5% non-fat dry milk) and washed, the protein was detected by incubating overnight at 4 °C with specific primary antibodies, as listed above. As secondary antibodies, we used HRP-conjugated goat anti-mouse or anti-rabbit antibodies (diluted 1:10000), incubated at room temperature for one hour. Chemiluminescence signal was detected using LAS4000 Image Reader (Fujifilm 2.11, Life Science) and quantified using the Fujifilm Multi Gauge software (V3.0). Signal for each sample of GRK2/3 was background-corrected and then normalized to its respective background-corrected actin signal.

### Localization of GRK constructs using confocal microscopy

For the assessment of employed GRK construct localization, we utilized N-terminally tagged NeonGreen-GRK fusion protein plasmids. Corresponding to the respective construct, a C-terminal H-Ras CAAX motif tethers the GRK to the plasma membrane. The cells were seeded in 6 cm dishes (ΔQ-GRK $1.6 \times 10^6$ cells per dish) and transfected the following day with 1 µg of the indicated NeonGreen-GRK construct or empty vector (EV), according to the Effectene transfection reagent manual (Qiagen, #301427). After 24 h $1.0 \times 10^6$ cells were re-seeded on poly-D-Lysine-coated 24 mm round glass coverslips and maintained for 24 h in complete culture medium. Images of living cells were acquired using an inverted laser scanning confocal microscope (DMi8 TCS SP8, Leica microsystems) equipped with a HC PL APO CS2 63x/1.40 oil objective (Leica).The NeonGreen fluorophore was excited at a wavelength of 496 nm and emission was detected in a bandwidth of 512–540 nm. $1024 \times 1024$ pixels format images were acquired and later processed with ImageJ (https://imagej.nih.gov/ij/ NIH, Bethesda).

### Fluorometric assessment of GRK2 and GRK3 construct expression

ΔQ-GRK cells were seeded and transfected as described in above in the "Localization of GRK constructs using confocal microscopy" section. The following day, 40,000 cells per well were seeded into black poly-D-lysine-coated 96-well plates (Brand, 781968). For each transfection, technical replicates were seeded as quadruplicates. Before measuring the next day, the cells were washed twice using measuring buffer (140 mM NaCl, 10 mM HEPES, 5.4 mM KCl, 2 mM $CaCl_2$, 1 mM $MgCl_2$; pH 7.3). The measurements were performed in measuring buffer using a Synergy Neo2 plate reader (Biotek), the Gen5 software (version 2.09) and a corresponding 485/20 excitation filter (BioTek, 1035014) and 516/540 emission filter cube (BioTek, 1035047). The measured intensity was normalized to background (empty vector-transfected control).

### Bioluminescence resonance energy transfer (BRET) measurements

The intermolecular BRET measurements to investigate β-arrestin recruitment were conducted as described before[16]. In short, cells were seeded into 6 cm dishes (ΔQ-GRK $1.6 \times 10^6$ cells per dish) and transfected the following day with 0.5 µg of the indicated GPCR-NLuc, 1 µg of Halo-tag-β-arrestin2 and 0.25 µg of one GRK construct or empty vector (EV), according to the Effectene transfection reagent manual (Qiagen, #301427). Each transfection was adjusted with EV to contain 2.5 µg total DNA. The following day, 40,000 cells per well were seeded into poly-D-lysine-coated 96-well plates (Brand, 781965) and the Halo-ligand was added (1:2,000; Promega, G980A). For each transfection, technical replicates were seeded as triplicates and a mock labeling condition was included without the Halo-ligand. Before measuring the next day, the cells were washed twice using measuring buffer (140 mM NaCl, 10 mM HEPES, 5.4 mM KCl, 2 mM $CaCl_2$, 1 mM $MgCl_2$; pH 7.3). After aspiration, the NLuc-substrate furimazine (Promega, N157B) in measuring buffer (1:3,500) was added. The measurements were performed in a Synergy Neo2 plate reader (Biotek), the Gen5 software (version 2.09) and a customized filter cube (fitted with a 555 nm dichroic mirror and a 620/15 bandpass filter). First, basal values were measured for 3 min, followed by addition of the indicated agonist and measurement of the stimulated values for 5 min. After the first measurement upon stimulation, four data points (43 sec intervals) were averaged for the concentration-response curves. The human b2AR was stimulated with isoproterenol (Iso; Sigma-Aldrich, I5627, dissolved in water). The human M2R and M5R were stimulated with Acetylcholine (ACh; Sigma-Aldrich, A6625, dissolved in measuring buffer).

In case of the bARK-CT-mediated inhibition of the GRK2-Gβγ interaction[32,36] and the measured effects on β-arrestin2 recruitment, $1.2 \times 10^6$ CRISPR/Cas9 HEK293 control cells were seeded into 6 cm dishes. The cells were transfected as described above with 0.5 µg of M5R-NLuc and 1 µg of Halo-tag-β-arrestin2, in addition to 0.5 µg or 1 µg of bARK-CT or 1 µg EV as a control. The total amount of transfected DNA was adjusted to 2.5 µg with EV, when necessary.

For the measurement of β-arrestin2 recruitment to M5R in presence of the guanine nucleotide dissociation inhibitor FR900359[38], ΔQ-GRK and CRISPR/Cas9 HEK293 Control cells (Control) were transfected and re-seeded as described above. On the day of the measurement, cells were pre-incubated in cell culture medium (DMEM, 10% fetal calf serum, 1% peni-cillin and streptomycin mixture) supplemented with 300 nM FR900359 (10 mM stock solution, solved in DMSO) for 1 h at 37 °C. Following pre-incubation, cells were washed twice with measuring buffer supplemented with 300 nM FR900359. Addition of furimazine substrate and measure-ments were then carried out in measuring buffer supplemented with 300 nM FR900359. The data points at 2–4 min after stimulation were averaged for the concentration-response curves.

To measure the effect of FR900359 on receptor internalization, ΔQ-GRK and CRISPR/Cas9 HEK293 Control cells were seeded as described above and transfected with 0.1 µg M5R-NLuc and 1 µg early endosome-tethered mNeonGreen-FYVE[39–41]. The total amounts of DNA were adjusted to 2 µg and the transfections were carried out according to the Effectene transfection reagent manual (Qiagen, #301427). The pre-incubation with 300 nM FR900359 was performed as described above. The measurement was performed using a 410/515 filter (BioTek, 1035072) for 30 min after stimulation. The data points of the last 10 min were averaged for the concentration-response curves.

### Analysis, statistics and reproducibility

The measured BRET ratios were labeling corrected by subtraction of the respective mock-labelled condition and subsequently the averaged technical replicates of the stimulated values were divided by the respective averaged baseline values. To get the final dynamic Δ net BRET change, the ligand-dependent labeling-corrected BRET change was divided by the vehicle control and calculated as percent changes. These corrected BRET changes were normalized to the maximum value of GRK2- or GRK-CAAX-mediated recruitment, as indicated in the respective figure legends. In the bARK-CT experiment, the Δ net BRET changes were normalized to the β-arrestin2 recruitment at the highest ligand concentration in absence of bARK-CT (EV-transfected condition). For the FR900359 inhibition experiments, the Δ net BRET changes were normalized to the maximal signal in CRISPR/Cas9 HEK293 control cells without the inhibitor present. In case of the M5R translocation to early endosomes, no labeling procedure was necessary due to utilization of the fluorophore NeonGreen as acceptor, hence there was no labeling correction in the analysis. All data are shown as mean of at least three independent experiments ± SEM as indicated. Sta-tistical comparisons were made in GraphPad Prism 7.03 using one-way ANOVA and subsequent Turkey's test. The supplementary bar graphs of the Halo labeling- and vehicle-corrected mean Δ net BRET changes + SEM before (basal) and after stimulation with the indicated ligand were nor-malized to the basal BRET ratio derived from the EV-transfected condition (Δ net BRET fold change). Here, statistical differences within one condition between basal and stimulated or between differently transfected conditions were tested using two-way ANOVA followed by a Sidak's or Tukey's test respectively. In all cases, a type I error probability of 0.05 was considered significant.

Plate reader experiments were performed in technical replicates of three or four from the same transfection, stimulated with identical ligand solution, as indicated in the respective methods sections. The technical replicates were averaged for each $n$. All shown experi-ments represent at least $n = 3$ of independent experiments with inde-pendent transfections. Exact $n$ numbers are provided in the respective figure legends.

### Data availability

All source data supporting the findings of this work presented in the main Figures are available within the article and supplementary information files. The numerical source data behind the main figures are available in Sup-plementary Data 1–5. All data displayed in the Supplementary Figures are available from the corresponding author on reasonable request.

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

## Acknowledgements

We want to thank Prof. Silvio Gutkind for kindly providing the bARK-CT plasmid and Prof. Ignacio Rubio for insightful discussions about membrane-tethering systems. Additionally, the authors would like to thank Nadia Beaupain for technical support. This work was supported by the European Regional Development Fund (Grant ID: EFRE HSB 2018 0019) and by ONCORNET2.0 - H2020-MSCA-ITN-2019; Grant Agreement number: 860229 to C.H. This study was additionally supported by the Luxembourg Institute of Health (LIH) through the NanoLux platform, Luxembourg National Research Fund (INTER/FNRS grants 20/15084569 and CORE C23/BM/18068832) and F.R.S.-FNRS-Télévie (grant 7.4547.19). A.G. and J.D. were supported by the University Hospital Jena IZKF (Grant ID: MSP 11 and MSP 10). N.J. was funded with a PhD fellowship from the IZKF, University Hospital Jena. J.C.F. was supported by the Jena School of Molecular Medicine funded by the Carl-Zeiss-Stiftung. E.K. gratefully acknowledges DFG funding support for the grant 290847012/FOR2372.

## Author contributions

C.H. developed the concept of the study. E.S.F.M., J.C.F., N.J., N.Y., M.R., and A.G. performed and analyzed all measurements. G.D., M.S., and A.C. generated all mNeonGreen-coupled GRK constructs. J.D. engineered the GRK knockout cells. G.M.K., E.K. and A.C. provided material. E.S.F.M., M.R., A.G., N.Y., J.C.F. and C.H. wrote the manuscript with contributions and critical feedback from all authors.

## Funding

## Competing interests
