## [Peer review file · Communications Biology]

Reviewers' comments:

Reviewer #1 (Remarks to the Author):

The authors provide a systematic assessment for which GRKs are required for the recruitment of arrestins to GPCRs. The main conclusion is that some receptors require (exclusively) G proteins ($G_{\beta\gamma}$) dependent action of GRK2/3 family GRKs, whereas others also work through another family of GRKs (GRK4/5/6). The results are not entirely surprising, but the work is a much needed addition to the field in the form of a systematic and definitive study. The experiments are well done, the quality of the data is high, and the conclusions are well supported for the most part. I only have minor suggestions to tone down or qualify one of the conclusions stated in the abstract and in several passages of the manuscript.

1- The authors make a categorical conclusion in the abstract: "we propose that it will likely be mechanistically unattainable to create β -arrestin-biased ligands for the subgroup of GRK2/3-regulated GPCRs".

This might be true for "pure" arrestin biased ligands, but ligand bias is a continuum. Based on the evidence presented, it cannot be completely ruled out that a ligand with weak G protein activating capacity could lead to enhanced arrestin recruitment/ signaling because the thresholds of different events might not relate linearly. For example, a weak G protein activation may still lead to full recruitment of arrestins (or recruitment of the signaling "active" conformation of arrestins) if GRK phosphorylation leads to amplification. I would agree that it will be more challenging to find arrestin biased ligands for GRK2/3 dependent GPCRs (there is less room to find the desired behavior for a particular compound).

I will list next some passages that I think should be revised in addition to the abstract to address this point:

>>> "Hence, we would like to broadly classify GPCRs based on their GRK-selectivity and connect this classification to a consequent possibility of creating biased agonists (Fig. 6)." --- I would replace "possibility" by another word. Maybe tractability?

>>> "The creation of β -arrestin-biased agonists would likely be mechanistically unattainable for GRK2/3-regulated GPCRs, such as the M2R or M5R, as the GRK2/3-mediated β -arrestin effects are indeed $G_{\beta\gamma}$ -dependent." --- I think here the authors should expand the discussion to distinguish pure arrestin biased ligands versus other possibilities like what I describe above related to the abstract.

>>> "Ultimately, our findings hope to convey the simple message that we should consider whether there is a "GRK bias" of a receptor to assess its potential in G protein- and β -arrestin-biased signaling." --- I would again consider to replace possibility by another word.

2- Change the title to "GRK specificity and $G_{\beta\gamma}$ dependency determines a GPCR's potential for arrestin biased agonism" for the sake of accuracy

3- Really minor, I would refrain from referring to "our" cells, especially in the abstract. I agree that they were generated by the investigator, but that does not seem very relevant for the scientific message.

Reviewer #2 (Remarks to the Author):

The manuscript by Matthees et al examines the role of the GRK2/3 interaction with $G_{\beta\gamma}$ in the GRK2/3-dependent regulation of the signaling via G protein-coupled receptors (GPCR). The interaction with $G_{\beta\gamma}$ is the means of recruiting cytosolic GRK2/3 to the plasma membrane, and $G_{\beta\gamma}$ has been known to facilitate GRK2/3-dependent GPCR phosphorylation. Here the authors argue that the dependence of the GRK2/3 recruitment to the membrane on $G_{\beta\gamma}$ makes the G protein activation indispensable for the GPCR phosphorylation by GRK2/3 and subsequent arrestin

recruitment.

Page 4, line 115: The sentence is unclear: "As depicted in Fig 1a, these mutant versions" – which mutants? Furthermore, here in the text it appears that the mutations are in G protein subunits – shown as $G\alpha_q$ (D110A) and $G\beta\gamma$ (R587Q) – whereas in fact they are in GRKs2/3 as is evident from Methods as well as Fig 1. Please, clarify.

It is surprising that no expression data are provided for the GRK mutants. The authors are certainly well aware that mutation affect the expression of proteins in an unpredictable way. To enable meaningful comparison, the expression levels need to be equalized across experiments.

Additionally, since the issue is the membrane localization, the subcellular localization of the GRK mutants should be documented. The CAAX motive is a powerful membrane attachment signal. Nevertheless, it should be experimentally demonstrated that all GRK mutants equipped with the CAAX motive show comparable membrane localization.

The data with GRK3 are less clear-cut than with GRK2. A convincing rescue of the double mutant by CAAX was achieved only for M5R but not for M2R or b2R (no significant difference from control). In the latter two cases, it is hard to be sure, for no direct comparison between GRK3(D110A, R587Q) and GRK3(D110A, R587Q)-CAAX has been performed. Judging by the appearance, though, there does not seem to be any significant rescue.

An interesting question is why it should be harder to rescue a double GRK3(D110A, R587Q) mutant than a single GRK3(R587Q) mutant. If the issue were just the membrane recruitment, there should be no difference.

Also, the differences between GRK2 and GRK3 are glossed over in the discussion, but this is an interesting point. The two GRKs are very similar in structure and biochemically, and yet GRK3 tends to be more membrane-associated than GRK2 in many cells. It would be interesting to see the effect of mutations on its subcellular localization.

The authors claim that "GRK2/3-mediated β -arrestin2 recruitment to activated receptors will always be preceded by G protein activation". They also claim that there is a "GRK dependence on free $G\beta\gamma$ ". This is a very strong claim, and the data do not quite live up to it. The data support the notion that the GRK2/3 interaction with $G\beta\gamma$ promotes GRK2/3-dependent receptor phosphorylation and arrestin recruitment. It is important to note, however, that this has been known for some time – see, for example this study from Benovic's lab:(Kim et al. 1993).

The Discussion appears somewhat unfocused. Although there might be implications of the findings for the biased signaling, the biased signaling is not the focus of the study. The issue of GRK selectivity of GPCR was brought up but not really discussed, which appears more relevant to the study. Most GPCRs are phosphorylated by both GRK2/3 and GRK5/6 classes. It would be of interest to discuss the potential functional consequences of the dependence of one class of G protein activation and lack of that dependence of the other, and what the final outcome could be.

Kim, C. M., Dion, S. B. and Benovic, J. L. (1993) Mechanism of beta-adrenergic receptor kinase activation by G proteins. *J Biol Chem* 268, 15412-15418.

Reviewer #3 (Remarks to the Author):

In this paper, Matthees et al reported that GRK2/3 phosphorylation of several GPCRs and the subsequent β -arrestin2 coupling requires free $G\beta\gamma$, therefore they concluded that arrestin bias is less likely achieved for the subgroup of GRK2/3-regulated GPCRs. It is interesting and potentially

important for developing biased agonists; however, the authors don't take into account that there are different sources of free Gbg. For instance, one agonist could activate several GPCRs and the free Gbg can be made available via activating any receptor that responds to this agonist. One good example is from the Schafer et al 2023 Mol. Pharm. paper. The authors reported that the activation of CXCR4 provides free Gbg to boost the GRK2/3 mediated phosphorylation of ACKR3 and subsequent arrestin binding even though ACKR3 does not activate G protein itself. The authors need to at least address this in the discussion.

For the BRET assay, the authors compared GRK2 and GRK2-CAAX variants separately. Does GRK2-CAAX boost the BRET between b-arrestin2 and receptor by tethering to the membrane? The comparison between GRK2 and GRK2-CAAX is lacking.

Point-to-point reply to the reviewers' comments:

We thank all reviewers and the editor for their constructive feedback and helpful suggestions regarding the manuscript at hand. In the revised version of this manuscript, we addressed the points raised by the referees. We are convinced that this improved our existing work and manuscript text.

Reviewer #1 (Remarks to the Author):

The authors provide a systematic assessment for which GRKs are required for the recruitment of arrestins to GPCRs. The main conclusion is that some receptors require (exclusively) G proteins (Gbg) dependent action of GRK2/3 family GRKs, whereas others also work through another family of GRKs (GRK4/5/6). The results are not entirely surprising, but the work is a much needed addition to the field in the form of a systematic and definitive study. The experiments are well done, the quality of the data is high, and the conclusions are well supported for the most part. I only have minor suggestion to tone down or qualify one of the conclusion stated in the abstract and in several passages of the manuscript.

We thank the reviewer for their positive assessment of our study. We hope that the following point-to-point reply adequately addresses the questions and suggestions.

1- The authors make a categorical conclusion in the abstract: "we propose that it will likely be mechanistically unattainable to create β -arrestin-biased ligands for the subgroup of GRK2/3-regulated GPCRs".

This might be true for "pure" arrestin biased ligands, but ligand bias is a continuum. Based on the evidence presented, it cannot be completely rule out that a ligand with weak G protein activating capacity could lead to enhanced arrestin recruitment/ signaling because the thresholds of different events might not relate linearly. For example, a weak G protein activation may still lead to full recruitment of arrestins (or recruitment of the signaling "active" conformation of arrestins) if GRK phosphorylation leads to amplification. I would agree that it will be more challenging to find arrestin biased ligands for GRK2/3 dependent GPCRs (there is less room to find the desired behavior for a particular compound).

Inspired by the reviewer's insightful elaboration, we revised the respective sentence in the abstract to express it as "mechanistically challenging". To adhere to the journal's abstract length constraints, we have provided additional detail only in the sentences highlighted by the referee below.

I will list next some passages that I think should be revised in addition to the abstract to address this point:

>>> "Hence, we would like to broadly classify GPCRs based on their GRK-selectivity and connect this classification to a consequent possibility of creating biased agonists (Fig. 6)." --- I would replace "possibility" by another word. Maybe tractability?

In response to the referee's feedback, we revised the sentence in the discussion to include "consequential strategy to create β -arrestin-biased agonists (Fig. 6)", instead of "possibility". In agreement with the raised points by the referee above, evaluating the GRK specificity of a GPCR is crucial in determining the most effective approach for developing biased agonists tailored to the receptor of interest. As the reviewer correctly highlights, the thresholds of different GPCR signaling and regulatory events may not necessarily correlate in a linear fashion. However, the specific strategies to influence this remain yet to be determined. We hope to convey this more effectively with the revised version.

>>>"The creation of β -arrestin-biased agonists would likely be mechanistically unattainable for GRK2/3-regulated GPCRs, such as the M2R or M5R, as the GRK2/3-mediated β -arrestin effects are indeed $G\beta\gamma$ -dependent."--- I think here the authors should expand the discussion to distinguish pure arrestin biased ligands versus other possibilities like what I describe above related to the abstract.

We thank the reviewer for highlighting this discussion point. To address this, we revised the sentence to specifically refer to pure β -arrestin-biased agonists, which would be mechanistically challenging to create. Additionally, we inserted two sentences to incorporate the referee's suggestion, mentioned above, when talking about the abstract. Now this part reads: "The creation of pure β -arrestin-biased agonists would be mechanistically challenging for GRK2/3-regulated GPCRs, such as the M2R or M5R, as the GRK2/3-mediated β -arrestin effects are indeed $G\beta\gamma$ -dependent. Partial agonists could lead to G protein-biased signaling if the subsequently available $G\beta\gamma$ is not sufficient to mediate efficient GPCR phosphorylation by GRK2/3. However, the thresholds for these mechanisms remain unknown. If activation of and phosphorylation by GRKs exhibit some degree of amplification, it could also be imaginable that weak G protein activation may still result in comparable β -arrestin recruitment levels." We hope that this strengthens the points the referee raised.

>>>"Ultimately, our findings hope to convey the simple message that we should consider whether there is a "GRK bias" of a receptor to assess its potential in G protein- and β -arrestin-biased signaling." --- I would again consider to replace possibility by another word.

We changed the last sentence of the discussion to "Ultimately, our findings hope to convey the importance of evaluating whether a receptor exhibits a "GRK bias" to assess optimal strategies for inducing G protein- or β -arrestin-mediated cellular responses." Through these adjustments, we aim to articulate the key take-away message in a clearer way, without overstating it. We hope these changes sufficiently address the referee's concerns.

2- Change the title to "GRK specificity and $G\beta\gamma$ dependency determines a GPCR's potential for arrestin biased agonism" for the sake accuracy

We appreciate the reviewer's suggestion. We incorporated "arrestin" into the title to specifically refer to arrestin-biased agonism. This adjustment enhances the accuracy of the title in representing the focus of the study.

3- Really minor, I would refrain from referring to "our" cells, especially in the abstract. I agree that they were generated by the investigator, but that does not seem very relevant for the scientific message.

We agree with the reviewer's observation that the origin of the utilized GRK2/3/5/6 knockout cells, generated in the same laboratory and by the authors conducting this study, is not pertinent to the scientific message conveyed in this manuscript. Our usage of this reference was merely habitual. In response, we revised the relevant sentences throughout the manuscript in a more neutral way.

Reviewer #2 (Remarks to the Author):

The manuscript by Matthees et al examines the role of the GRK2/3 interaction with G $\beta\gamma$ in the GRK2/3-dependent regulation of the signaling via G protein-coupled receptors (GPCR). The interaction with G $\beta\gamma$ is the means of recruiting cytosolic GRK2/3 to the plasma membrane, and G $\beta\gamma$ has been known to facilitate GRK2/3-dependent GPCR phosphorylation. Here the authors argue that the dependence of the GRK2/3 recruitment to the membrane on G $\beta\gamma$ makes the G protein activation indispensable for the GPCR phosphorylation by GRK2/3 and subsequent arrestin recruitment.

We thank the reviewer for the comprehensive summary and fair review of our study. In the following we elaborate on the changes that aim to resolve the specified suggestions and questions raised by the referee.

Page 4, line 115: The sentence is unclear: "As depicted in Fig 1a, these mutant versions" – which mutants? Furthermore, here in the text it appears that the mutations are in G protein subunits – shown as G α q(D110A) and G $\beta\gamma$ (R587Q) - whereas in fact they are in GRKs2/3 as is evident from Methods as well as Fig 1. Please, clarify.

We apologize for the unclear wording in the sentence. The term "these mutant versions" referred to the previous sentence, in which we mention GRK2-mutants with low binding affinity towards the G protein subunits. As correctly pointed out by the reviewer, these mutations affect GRK2/3, not the G proteins themselves. Therefore, we have revised the sentence as follows: "As depicted in Fig. 1a, these mutations in GRK2 disrupt its interaction with G α q (GRK2-D110A), its interaction with G $\beta\gamma$ (GRK2-R587Q) or both (double mutation at D110A and R587Q in GRK2)." We hope this clarification resolves any ambiguity in the sentence.

It is surprising that no expression data are provided for the GRK mutants. The authors are certainly well aware that mutation affect the expression of proteins in an unpredictable way. To enable meaningful comparison, the expression levels need to be equalized across experiments.

We thank the referee for raising this important issue and apologize at the same time. Concurrently, we have been engaged in another manuscript (Jaiswal et al. in preparation: "New insights into the non-canonical desensitization of Gq-signaling by GRK2/3 expression levels"; a meeting Abstract of this work can be found at <https://www.webofscience.com/wos/woscc/full-record/WOS:000765769800159>) that also involves these GRK mutants. The characterization of the utilized mutants, like investigating their expression, has been incorporated into that manuscript since it was supposed to be published first but got delayed. We are close to finishing that manuscript as well to ensure accessibility for both the referee and the possible readers (Jaiswal et al., in preparation). For the referees, we provided parts of that material here in the reply file to enable a convenient comparison (Fig. 1). Meanwhile, we are continuing our efforts towards publication of that work in a peer-reviewed journal. We provide the direct reference "as in preparation" in the results section, where the mutants are initially mentioned, and in the corresponding methods section.

Figure 1: Western blot analysis of cells expressing GRK2 or 3 wild type or mutant constructs, unable to interact with G protein subunits, against actin as a loading control (Jaiswal *et al.*, *in preparation*).

Additionally, since the issue is the membrane localization, the subcellular localization of the GRK mutants should be documented. The CAAX motive is a powerful membrane attachment signal. Nevertheless, it should be experimentally demonstrated that all GRK mutants equipped with the CAAX motive show comparable membrane localization.

We appreciate the referee's suggestion regarding the documentation of the subcellular localization of the GRK-CAAX mutants. In response, we employed fluorophore-coupled versions of the GRK-CAAX constructs and acquired confocal images, as well as intensity measurements. This documentation is now presented in a newly created Supplementary Figure 1 and 5 and detailed in the methods section ("Localization of GRK constructs using confocal microscopy", "Fluorometric assessment of GRK2 and GRK3 construct expression"). The statistical comparisons can be accessed in Supplementary Table 1 and 8. We trust that this satisfyingly resolves the reviewer's concern.

The data with GRK3 are less clear-cut than with GRK2. A convincing rescue of the double mutant by CAAX was achieved only for M5R but not for M2R or b2R (no significant difference from control). In the latter two cases, it is hard to be sure, for no direct comparison between GRK3(D110A, R587Q) and GRK3(D110A, R587Q)-CAAX has been performed. Judging by the appearance, though, there does not seem to be any significant rescue. An interesting question is why it should be harder to rescue a double GRK3(D110A, R587Q) mutant than a single GRK3(R587Q) mutant. If the issue were just the membrane recruitment, there should be no difference. Also, the differences between GRK2 and GRK3 are glossed over in the discussion, but this is an interesting point. The two GRKs are very similar in structure and biochemically, and yet GRK3 tends to be more membrane-associated than GRK2 in many cells. It would be interesting to see the effect of mutations on its subcellular localization.

We agree with the referee that even though the tendency was comparable, the observed differences between the obtained results with GRK2 or GRK3 raise a compelling point. An increasing body of evidence supports the notion that indeed GRK2 and GRK3 are not redundant. To emphasize this further, we have integrated it in the beginning of the discussion: "Analogously, we showed that this mechanism is also conserved for GRK3 across all investigated GPCRs (Fig. 3), albeit the clarity of these results interestingly appears to be somewhat less pronounced than for GRK2." With the addition of the CAAX motif, the membrane localization and hence independence of the G protein subunit interaction for the

recruitment are ensured. However, the exact complex geometry might differ compared to the wild type GRKs and the interaction with G $\beta\gamma$ could potentially play an additional role in mediating a specific complex configuration. To specifically confirm the GRK localization, we have included confocal images of each utilized GRK construct, coupled to a fluorophore (NeonGreen) and fluorescence intensity measurements to assess comparable expression in the revised version of the manuscript (new Supplementary Figure 1 and 5). Interestingly, all mutated constructs were cytosolic or membrane-localized, as expected based on whether the CAAX motif was added or not, similarly for GRK2 and GRK3. Moreover, statistical comparison revealed no significant differences in the intensity measurements between the wild type and CAAX constructs, as well as the mutant version (Supplementary Table 1 and 8).

The authors claim that “GRK2/3-mediated β -arrestin2 recruitment to activated receptors will always be preceded by G protein activation”. They also claim that there is a “GRK dependence on free G $\beta\gamma$ ”. This is a very strong claim, and the data do not quite live up to it. The data support the notion that the GRK2/3 interaction with G $\beta\gamma$ promotes GRK2/3-dependent receptor phosphorylation and arrestin recruitment. It is important to note, however, that this has been known for some time – see, for example this study from Benovic’s lab (Kim et al. 1993).

Kim, C. M., Dion, S. B. and Benovic, J. L. (1993) Mechanism of beta-adrenergic receptor kinase activation by G proteins. *J Biol Chem* 268, 15412-15418.

We concur with the reviewer’s observation regarding the established significance of the G $\beta\gamma$ –GRK2/3 interaction for the cytosolic GRKs to be activated and recruited to the membrane. This aspect is further highlighted by the reference provided by the referee here, which we have now additionally incorporated into the present manuscript. Nevertheless, we believe that its implications for receptor regulation, particularly concerning GRK2/3-regulated GPCRs, in the context of developing arrestin-biased ligands, have not received the warranted attention to date.

The Discussion appears somewhat unfocused. Although there might be implications of the findings for the biased signaling, the biased signaling is not the focus of the study. The issue of GRK selectivity of GPCR was brought up but not really discussed, which appears more relevant to the study. Most GPCRs are phosphorylated by both GRK2/3 and GRK5/6 classes. It would be of interest to discuss the potential functional consequences of the dependence of one class of G protein activation and lack of that dependence of the other, and what the final outcome could be.

We appreciate the referee’s evaluation of our discussion section. In response, we have introduced additional line breaks to enhance the visual organization and coherence of our discussion text, facilitating clearer paragraph structures. With this, we aim to more clearly structure the discussion into the following sections: a summary of our findings, what the GRK-specificity could mean for the effort to create β -arrestin-biased agonists for GRK2/3-, GRK5/6-, GRK2/3/5/6-regulated receptors and independent of GRK phosphorylation, followed by the outlook and take-home message. While we acknowledge the reviewer’s observation that biased signaling is not the primary focus of our study’s results, we maintain that our findings hold important implications for the development of biased agonists and which strategies are most promising to employ for different receptors. Therefore, we believe it is pertinent to discuss our results within this framework. We agree with the referee’s observation regarding the significant interest in exploring functional consequences of varying GRK2/3 or GRK5/6 dependencies, especially for receptors that can be regulated by all four ubiquitously expressed GRK isoforms. This aspect warrants thorough further investigation, as highlighted in our discussion section as well (e.g. “Finally yet importantly, as β -arrestin recruitment to GRK2/3/5/6-regulated GPCRs is partly G protein-dependent (GRK2/3) and partly independent (GRK5/6), the creation of β -arrestin-biased agonists targeting GRK2/3/5/6-regulated GPCRs is generally possible. However, these would only mediate β -arrestin effects linked to GRK5/6 phosphorylation while GRK2/3 effects would be lacking since the absence of G protein activation would not deliver free G $\beta\gamma$.”; “Hence, when designing β -arrestin-biased agonists one should keep in mind that the GRK5/6-mediated β -arrestin downstream

effects might differ substantially from the functional outcome facilitated by GRK2/3/5/6-recruited β -arrestin.”).

Reviewer #3 (Remarks to the Author):

In this paper, Matthees et al reported that GRK2/3 phosphorylation of several GPCRs and the subsequent β -arrestin2 coupling requires free G $\beta\gamma$, therefore they concluded that arrestin bias is less likely achieved for the subgroup of GRK2/3-regulated GPCRs.

We appreciate the referee’s concise summary of our study. In the following, we aim to satisfyingly respond to the raised points.

It is interesting and potentially important for developing biased agonists; however, the authors don't take into account that there are different sources of free G $\beta\gamma$. For instance, one agonist could activate several GPCRs and the free G $\beta\gamma$ can be made available via activating any receptor that responds to this agonist. One good example is from the Schafer et al 2023 Mol. Pharm. paper. The authors reported that the activation of CXCR4 provides free G $\beta\gamma$ to boost the GRK2/3 mediated phosphorylation of ACKR3 and subsequent arrestin binding even though ACKR3 does not activate G protein itself. The authors need to at least address this in the discussion.

We are grateful for the raised argument by the referee. In response, we included this point and the respective reference in the respective discussion paragraph, addressing GRK5/6-regulated receptors. The section now reads: “It has been shown for the atypical chemokine receptor ACKR3 that GRK2/3-mediated phosphorylation can be induced by overexpression of G $\beta\gamma$ and delivery of free G $\beta\gamma$ in the receptor vicinity via hetero-dimerization with the G protein-activating CXCR4, if stimulated by CXCL12 as a shared agonist for both receptors (Schafer et al. 2023, Mol Pharmacol). In this case, the same ligand induced active conformations of the receptors, which might have enabled GRK2/3 to phosphorylate also the atypical ACKR3 even though the recruitment to the membrane was mediated via CXCR4.”

For the BRET assay, the authors compared GRK2 and GRK2-CAAX variants separately. Does GRK2-CAAX boost the BRET between β -arrestin2 and receptor by tethering to the membrane? The comparison between GRK2 and GRK2-CAAX is lacking.

We thank the referee for raising this important point, which also intrigued us. Consequently, we conducted a direct comparison of the wild type GRK2/3 with their respective GRK2/3-CAAX versions regarding their capability to mediate β -arrestin recruitment to the b2AR, M2R and M5R (formerly Supplementary Figure 1-4, now Supplementary Figure 2-4, 6, with the inclusion of a localization control for the CAAX versions in the new Supplementary Figure 1 and 5). Interestingly, we observed that tethering the GRK2 to the membrane did not impact its ability to facilitate β -arrestin recruitment to the b2AR. However, it did result in an increase in the agonist-independent interaction of β -arrestin with the M2R and M5R. Notably, this effect was observed only for the M5R in case of GRK3.

REVIEWERS' COMMENTS:

Reviewer #1 (Remarks to the Author):

The authors have addressed my previous (minor) concerns. I congratulate the authors on the excellent work.

Note: regarding my last point about the use of "our", just wanted to reiterate that it was a very minor issue, and that I completely get that "Our usage of this reference was merely habitual." We probably do the same from time to time. Habits die hard.

Reviewer #2 (Remarks to the Author):

The authors have adequately answered the main concerns. Only one point remains: it is unclear why the GRK mutant expression data cannot be presented in this paper, at least, in the Supplement. It seems a little odd to refer to another manuscript, published or unpublished, for a rather important piece of control data.

Reviewer #3 (Remarks to the Author):

The authors have sufficiently addressed my comments. Nice to see the direct comparison between the GRK2/3 and GRK2/3-CAAX. My guess is that the membrane recruitment and receptor-mediated kinase activation have different levels of influence on arrestin binding depending on the GRK isoform and receptor type. Nice work!

Point-to-point reply to the reviewers' and editors' comments:

We are grateful for the final comments of the reviewers and editors and that we were given the chance to improve the manuscript on last time.

In the following, we will address the final points raised and hope this will enable us to publish the manuscript.

Reviewer #1 (Remarks to the Author):

The authors have addressed my previous (minor) concerns. I congratulate the authors on the excellent work.

Note: regarding my last point about the use of "our", just wanted to reiterate that it was a very minor issue, and that I completely get that "Our usage of this reference was merely habitual." We probably do the same from time to time. Habits die hard.

We thank the reviewer for their understanding and positive feedback. We will keep this in mind for our future work and hope we are not going for the sequel "die harder"...

Reviewer #2 (Remarks to the Author):

The authors have adequately answered the main concerns.

Only one point remains: it is unclear why the GRK mutant expression data cannot be presented in this paper, at least, in the Supplement. It seems a little odd to refer to another manuscript, published or unpublished, for a rather important piece of control data.

We appreciate the reviewer's concern. To address this, we included a new Suppl. Fig. 1 and methods section to provide Western blot data for the GRK constructs. Additionally, we cite Natasha Jaiswal's dissertation to provide additional information. However, now the Western blots are also provided in the manuscript at hand. We trust that this satisfyingly resolves the reviewer's concern.

Reviewer #3 (Remarks to the Author):

The authors have sufficiently addressed my comments. Nice to see the direct comparison between the GRK2/3 and GRK2/3-CAAX. My guess is that the membrane recruitment and receptor-mediated kinase activation have different levels of influence on arrestin binding depending on the GRK isoform and receptor type. Nice work!

We are grateful for the reviewer's work and positive assessment of our revised work.